# Predictors of Smoking in Older Adults and an Epigenetic Validation of Self-Report

**DOI:** 10.3390/genes14010025

**Published:** 2022-12-22

**Authors:** Jeffrey D. Long, Michael P. Gehlsen, Joanna Moody, Gracie Weeks, Robert Philibert

**Affiliations:** 1Department of Psychiatry, Carver College of Medicine, University of Iowa, Iowa, IA 52242, USA; jeffrey-long@uiowa.edu (J.D.L.); joanna-moody@uiowa.edu (J.M.); gracie-weeks@uiowa.edu (G.W.); 2Department of Biostatistics, College of Public Health, University of Iowa, Iowa, IA 52242, USA; 3South Saint Paul Public Schools, 104 5th Ave. S, South Saint Paul, MN 55075, USA; gehlsen.michael@gmail.com; 4Behavioral Diagnostics LLC, 2500 Crosspark Rd., Suite W245, Coralville, IA 53341, USA; 5Department of Biomedical Engineering, University of Iowa, Iowa, IA 52242, USA

**Keywords:** smoking and body weight, self-report smoking, National Lung Screening Trial, cg05575921 methylation

## Abstract

There are several established predictors of smoking, but it is unknown if these predictors operate similarly for young and old smokers. We examined clinical data from the National Lung Screening Trial (NLST) to determine the predictive ability of gender, body mass index (BMI), marital status, and race on smoking behavior, with emphasis on gender interactions. In addition, we validated the self-report of smoking behaviors for a subgroup that had available epigenetic data in the form of cg05575921 methylation. Participants were 
N=9572
 current or former smokers from the NLST biofluids database, age 55–74, minimum of 30 pack years, and mostly White. A subgroup of 
N=3084
 who had DNA were used for the self-report validation analysis. The predictor analysis was based on the larger group and used penalized logistic regression to predict the self-report of being a former or current smoker at baseline. Cg05575921 methylation showed a moderate ability to discriminate among former and current smokers, AUC = 0.85 (95% confidence interval = [0.83, 0.86]). The final selected variables for the prediction model were BMI, gender, BMI by gender, age, divorced (vs. married), education, and race. The gender by BMI interaction was such that males had a higher probability of current smoking for lower BMI, but this switched to females having higher current smoking for overweight to obese. There is evidence that the self-reported smoking behavior in NLST is moderately accurate. The results of the primary analysis are consistent with the general smoking literature, and our results provide additional specificity regarding the gender by BMI interaction. Body weight issues might play a role in smoking cessation for older established smokers in a similar manner as younger smokers. It could be that women have less success with cessation when their BMI increases.

## 1. Introduction

Tobacco smoking continues to be a costly public health problem and the leading cause of preventable death [1,2]. Smoking is especially problematic in older adults as it can exacerbate new or existing chronic diseases and greatly complicate their management [3]. Older smokers are established smokers, and as such, they might have different motivations for smoking than younger people and might require different strategies for successful cessation [4]. Cessation is always advised as it can increase the remaining years of life, even if quitting is late [5].

There are several predictors of smoking that provide insight into factors that might be alterable for potential cessation strategies. Prevalence of smoking has been found to differ by marital status, with the highest prevalence typically for divorced people and the lowest for those who are married [6,7]. Marriage is also associated with more successful smoking cessation [8]. The marriage effect is thought to be due to the increased economic, social, and psychological resources afforded by having a spouse [9]. The effect might be enhanced for older adults, as one’s spouse becomes a primary focus of daily life as couples age [10]. Conversely, divorced people can be more prone to isolation and reduced physical activity, which in turn might facilitate smoking as a coping mechanism [11,12,13].

Another important predictor of smoking is body weight. There tends to be an inverse relationship between body mass index (BMI) and smoking [14,15,16], and people with weight concerns are more likely to smoke [13,17]. Gender is a moderating factor, as females more often report smoking to achieve weight loss and/or maintain their body weight [18]. Cessation strategies for women are more successful when weight gain issues are addressed [19,20]. In contrast, males tend to be more externally motivated to smoke and respond better to cessation strategies that involve (dis)incentives [21].

Smoking is also related to socio-economic status (SES) conditions, such as education, with higher prevalence of smoking associated with lower education levels [14]. Smoking also tends to vary by race, with certain non-Whites having higher prevalence [22].

A complication with the predictors of smoking is that their effects do not appear to be only additive. Several researchers have found that gender is an important moderator variable for smoking research, interacting with both body weight [16,17,23] and marital status [12,22]. It is important then, to either stratify results by gender or include gender interaction terms in modeling. It is the latter approach that we adopt and feature below.

In addition to selecting the predictors to include in an analysis and specifying their effects (e.g., interactions), the integrity of the smoking outcome variable is important. The majority of smoking studies rely on self-report of smoking behavior, which might be affected by response bias [24,25,26]. The accuracy of the self-report appears to be population dependent, with trends of under-reporting found in certain groups, such as pregnant women [24]. Results of studies designed to validate the self-report measures with biological readouts have been mixed, ranging from very poor agreement [27] to very good or excellent agreement [28,29].

Most validation studies have focused on the agreement between cotinine level and self-report of smoking [30]. Limitations of cotinine level are that it can only detect nicotine use within 48–72 h, and it cannot distinguish between sources of nicotine, such as smoking and nicotine replacement therapy [31]. These factors might make cotinine level less than ideal for use with established smokers, some of whom may have many years of cessation. Recent developments in epigenetics provide alternatives, such as cg05575921 methylation [32]. Cg05575921 methylation has been shown to reliably assess the amount that someone has smoked with a much wider time window for detection than cotinine level [33].

The current study proposes to add to the smoking literature in the following ways. We will use a large sample (*N* > 9000) of older established smokers (age 55–75, minimum 30 pack years) to examine predictors of self-reported smoking. Focus will be on gender interactions, especially the interaction of gender with BMI and marital status in predicting smoking. We also consider education level and race. Prior to the analysis of the correlates, we examine the relationship between self-report smoking and cg05575921 methylation on a subgroup for whom DNA was available.

## 2. Materials and Methods

### 2.1. Participants

There were 
N=9572
 participants from the National Lung Screening Trial (NLST) [34] who had biofluids in the American College of Radiology Imaging Network (ACRIN) biomarker repository [35]. The NLST is a multi-site randomized trial that was initiated in 2002 with the primary goal to determine the effect of different screening strategies on mortality from lung cancer. It enrolled over 50,000 high risk individuals ages 55–74 years, who were heavy current cigarette smokers or former smokers. Participants had to have at least 30 pack years of smoking history, and if former smokers, they had to have quit within the previous 15 years. A much smaller number (
N≈
 10,300) were accrued in the ACRIN biomarker repository, which was completed in 2004. Additional details can be found on the NIH-NCI website (https://www.cancer.gov/types/lung/research/nlst, accessed on 29 November 2022).

The analysis sample was less than the total in the ACRIN repository because participants were required to have complete data and to be classified in the education, marital status, and race categories that were used for the analysis. Some categories were omitted because of low frequencies. For race, the categories used were non-Hispanic “White” and “Black or African-American”. For marital status the categories were “Never married”, “Married or living as married”, “Widowed”, “Separated”, and “Divorced”. The education categories were the seven that could be ordered (“8th grade or less”, “9th–11th grade”, “High school graduate/GED”, “Post high school training, excluding college”, “Associate degree/some college”, “Bachelors Degree”, “Graduate School”). Only the baseline data was analyzed.

For the self-report smoking validation, there were 
N=3084
 participants whose DNA was available to determine cg05575921 methylation [36]. Cg05575921 methylation was determined using the Smoke Signature assay from Behavioural Diagnostics (Coralville, IA, USA) [31,37]. Briefly, a 3 μL aliquot sample of bisulfite converted DNA is first pre-amplified under stringent conditions. The resulting product is next diluted 1:3000. A 5 μL aliquot having approximately 10,000 “C” or “T” containing amplicons is then added to aliquots of droplet digital PCR master mix from Bio-Rad (California) and fluorescent dual-labeled primer probe sets specific for cg05575921, partitioned into droplet, then subjected to PCR amplification. Finally, a Bio-Rad QX-200 droplet counter is used to determine the number of droplets containing amplicons that have a “C” allele (methylated cytosine residue), a “T” allele (unmethylated cytosine), at least one of each, or neither. The variable is then calculated as the percent of methylation [37].

### 2.2. Statistical Methods

#### 2.2.1. Self-Report Smoking Validation

Validation of self-report smoking used cg05575921 methylation as the predictor, and cigsmok as the outcome, which was cigarette smoking status at time of randomization (0 = Former, 1 = Current). Because there are no consensus cutoffs for cg05575921 methylation, an optimal cut-point analysis was performed. Non-parametric bootstrapping with cross-validation [38] was used to find the value of cg05575921 methylation that maximized the product of sensitivity and specificity, known as the concordance probability [39]. The area under the receiver-operator characteristic (ROC) curve (AUC) was the primary index of performance. In this context, AUC is the probability that a randomly chosen current smoker and a randomly chosen former smoker are correctly rank ordered on cg05575921 methylation. Lower cg05575921 methylation indicates greater smoking, so that a randomly chosen current smoker should have a lower level than a randomly chosen former smoker. Bootstrap cross-validation was used to compute AUC and its 95% confidence interval (CI; the CI was asymmetric due to the non-parametric method). Sensitivity and specificity for the selected (fixed) optimal cut-point were computed in order to compare our results to other studies. The number of years quit was computed as age minus the reported age at quitting cigarettes (age quit). Values for the variable were whole numbers because age and age at quitting were only reported as whole numbers. Years quit was used to gain additional insights into the relationship between smoking status and cg05575921 methylation.

#### 2.2.2. Predictors of Smoking

The second and primary analysis was for the entire sample and used penalized logistic regression to predict smoking status. The predictors were marital status, BMI, race, education, marital status by gender, BMI by gender, and race by education. The categories of marital status were dummy coded, except for “Married or living as married”, which was the reference category (resulting in four dummy variables). Education was dummy coded as 0 = high school or less, and 1 = greater than high school. Race was dummy coded as 0 = “Black or African-American”, 1 = “White” (both non-Hispanic; counts were too low to include other categories). Gender was coded as 0 = male, 1 = female. In total there were 16 predictors, nine single effects (main effects) and six interactions.

With the large sample size and moderate number of predictors, there was a concern of over-fitting by selecting too many variables. To guard against this, a two-step approach was used. First, the minimax concave penalty (MCP) [40] was used in the penalized logistic regression to shrink the coefficients of useless predictors to 0. Second, we computed a type of marginal false discovery rate (mFDR) [41] and used a stringent criterion for final variable selection (mFDR < 0.001). Cross-validation was used to select the tuning parameter of the MCP regression, and to compute the overall fit indices of AUC and 
$R^{2}$
.

All analysis was performed with the R computing platform [42] (version 4.1.3). The ggplot2 [43] package was used for graphing, cutpointr [38] was used for the cut-point estimation, and ncvreg [44] was used for the penalized logistic regression and mFDR computation.

#### 2.2.3. Role of Funding Source

The study sponsor had no role in the study design, the collection, analysis, or interpretation of data, in the writing of the report, or in the decision to submit the paper for publication.

## 3. Results

Table 1 shows the descriptive statistics for key variables of the sample with cg05575921, the sample without cg05575921, and the combined sample used for the primary analysis. There was a relatively small frequency of African Americans (<
6%
) and separated people (<
2%
). A small number of self-reported current smokers did report an age of quitting, which caused the mean years quit to be greater than 0. The samples with and without cg05575921 were similar on the demographic variables.

### 3.1. Self-Report Smoking Validation

The ROC analysis resulted in AUC = 0.85 with 95% CI = [0.83, 0.86]. The optimal cg05575921 methylation cut-point was 57.19 [55.93, 59.11]. For the selected (fixed) cut-point, sensitivity = 0.76 [0.74, 0.79], and specificity = 0.80 [0.76, 0.83].

Figure 1 shows the distribution of cg05575921 methylation by smoking status. Panel A shows boxplots wrapped in violin plots for current and former smokers with optimal cut-point (solid line) and (asymmetric) 95% CI (dashed lines). As the figure shows, the distribution for current smokers was shifted downward relative to former smokers. There was extensive overlap of the distributions. A small number of self-reported former smokers had methylation values in the lowest range of current smokers (23.4 to 35). Conversely, some self-reported current smokers had values at the highest range of former smokers (80 to 90), which is in the range of life-long non-smokers [24].

Panel B shows the boxplots of cg05575921 methylation by smoking status and years quit. For both smoking groups, cg05575921 methylation tended to increase with years quit. There were 119 (8%) self-reported former smokers who had 0 years quit, and 14 (1%) current smokers who reported more than 0 years quit, the latter being attributable to measurement error. For 0 years quit, the current smokers’ distribution was shifted downward relative to the former smokers’ distribution. For current smokers who reported greater than 1 year quit, their cg05575921 methylation medians tended to be at or above those of former smokers.

### 3.2. Predictors of Smoking

The results of the penalized logistic regression are shown in Table 2. The rows are sorted by the 
Z
-value and the last column shows the final set of variables selected (an asterisk indicates a variable was selected). The BMI main effect showed a negative relationship with log odds of current smoking (
${\hat{\beta }}_{BMI}=$
 −0.10, Z = −22.65), as did age (
${\hat{\beta }}_{age}=$
 −0.05, Z = −12.57). Men had a greater log odds of smoking (
${\hat{\beta }}_{gender}=$
 −0.77, Z = −18.01), as did African-Americans (
${\hat{\beta }}_{race}=$
 −0.71, Z = −6.36), and those who were divorced (
${\hat{\beta }}_{divorced}=$
 0.50, Z = 9.57). The BMI by gender interaction was positive (
${\hat{\beta }}_{BMI\times G}=$
 0.03, Z = 17.51), indicating a greater log odds for women as BMI increased. The bootstrap cross-validation fit indices for the set of variables were AUC = 0.65 and 
$R^{2}=$
 0.07.

Figure 2 shows BMI stratified by gender, with CDC cutoffs for different weight classes (https://www.cdc.gov/healthyweight/assessing/bmi/index.html, accessed on 26 September 2022). The BMI by gender interaction is illustrated in Figure 3, along with the main effect for age (the quartiles are depicted), and divorced vs. married. Probability of current smoking is shown as a function of BMI by gender, paneled by marital status (columns) and age (rows). The curves are the predicted probabilities from the fitted model fixing race to White and education to high (above high school). The vertical dashed lines demarcate the areas from left to right of underweight, healthy, overweight, and obese (class of obesity is not depicted). Figure 3 shows that for smaller BMI, the probability of current smoking was higher for males, but for larger BMI the converse was true. The cross-over to the higher probability for women occurred in the overweight range for married people and the obesity range for those who were divorced.

## 4. Discussion

The results of our study, which examined older established smokers, are consistent with the general smoking literature. We found a greater probability of smoking for divorced people vs. married, African Americans vs. Whites, and those with less education vs. those with more education. We also confirmed the well-documented inverse relationship between smoking and BMI. What is unique about our findings is the specificity of the BMI by gender interaction, which showed a cross-over from a higher male probability of smoking to a higher female probability as BMI increased. Furthermore, we validated the self-reported smoking status and showed very high agreement with an epigenetic marker of the amount of smoking.

Our validation results provide evidence of the reliability of self-reported smoking status in the NLST study. Despite some probable misreporting (see Figure 1), cg05575921 methylation had good ability to distinguish among current and former smokers. This result is similar to lung cancer screening studies that used cotinine level to verify self-reported smoking status and showed a high association between the two [28,45]. However, direct comparison of our results with the other studies is complicated due to the latter using pre-defined fixed cutoffs for cotinine (e.g., 4.85 ng/mL).

Using a fixed cutoff for cotinine means that a single value for sensitivity and for specificity can be directly computed. Not having a defined cutoff for cg05575921 methylation, we considered the AUC over multiple values of sensitivity and specificity. We did compute single values of sensitivity and specificity, but this was after the optimal cg05575921 methylation cut-point was determined (fixed). These values are lower than the values reported in the lung cancer screening studies. On the other hand, our values are well within the typical range for general studies that used cotinine level to validate self-reported smoking status [30].

Previous work has shown that cg05575921 methylation is correlated with cotinine level when used to detect daily or other short-term smoking [24,31]. Given that the NLST sample had several people with many years of smoking cessation, it is unknown how well cotinine level might discriminate smoking status, and how well it might track with cg05575921 methylation. The answer of course could be found if cotinine level were determined for the people who contributed cg05575921 methylation in our analysis, but this is a topic for future research.

The importance of body weight and gender in smoking was confirmed by the results of our main analysis. Our findings for BMI are consistent with the larger literature that has found it to predict smoking behavior in both males and females [46], with cessation resulting in an increase [47]. For our sample of older adults, as BMI increased, cessation probably accounted for the decreasing probability of smoking for males and females alike. A general motivation for quitting as one ages is the emergence of chronic illness that may or may not be related to smoking. Smoking can exacerbate chronic conditions, complicate treatment, and lead physicians to more forcefully recommend quitting [3,5].

As for the gender differences in smoking, we found that these varied based on the BMI range. At the lower ranges of BMI (underweight to healthy) there was a higher probability of smoking for males. This is probably a reflection of the overall higher rate of smoking among American men [48]. At the higher ranges of BMI (overweight and obese) there was a higher probability of smoking for females. Previous studies have shown that obese females are more likely to smoke than obese males [17,19]. Furthermore, obese males tend to smoke less than their healthy-weight counterparts, whereas the opposite is true for females [16].

A possible explanation for the BMI by gender interaction is that males and females have different levels of success in quitting smoking. It appears that both men and women attempt to quit smoking equally often, but it has been estimated that women are 31% less successful at continuing abstinence [49]. Because BMI is a predictor of smoking behavior, it tends to increase for both males and females as they quit smoking. The higher rate for females at greater BMI values might reflect less success in maintaining smoking abstinence [50].

A possible contributing factor is the gender difference in the motivation to smoke. Women are more likely to report that they smoke as a means of weight control [51], and cessation strategies tend to be more successful when the issue of weight gain is addressed [52]. Men are much less likely to endorse weight loss as a reason for smoking [53], and cessation strategies can be successful without explicitly addressing weight gain [19,21]. Therefore, it could be that women might be more prone to return to smoking as their BMI increases under cessation attempts. This is perhaps bolstered by our finding regarding the transition to the greater rate of female smoking. The transition occurred in the overweight to obese range (depending on other factors; see Figure 3). It is in this range that body weight might become an especially salient concern.

There is evidence that smoking has a greater impact on women’s health in terms of increased morbidity and mortality [19,50]. The combination of obesity and excess body weight is especially troublesome, as it raises the risk for cardiovascular diseases, diabetes, and certain cancers [54]. Thus, efforts to develop new and improve existing cessation strategies for women are particularly important. Our work suggests that issues regarding weight loss and smoking cessation may be important to target in older established smokers as they are in younger smokers [19,47,49,50].

Beyond the BMI by gender interaction, the remaining predictors of the final set were main effects (single-variable effects). There was an additive decrease in log odds of smoking with age, education, and being White, and an additive increase for divorced people (relative to those who were married) (see Figure 3). The age effect is similar to what was mentioned for BMI, that health issues manifest with age and probably increase the motivation to quit smoking [55]. Past work has shown that higher education level is associated with a lower rate of smoking and increased cessation success, but the reason for this effect is not clear and may be tied to a larger set of socioeconomic factors [56]. Similarly, past studies have found that cigarette smoking has a differential impact on US race groups [48]. We did not find the race by education effect that has been found in some other studies [57], though the frequency of African Americans in our sample was very low (3–5%).

Regarding marital status, we confirmed previous findings that married people tend to smoke less [6,22]. There was a significantly higher rate of current smoking for those who were divorced, but not for the other non-married categories. Divorce has consistently been shown to be a predictor of smoking, which might be due to a lack of support, a paucity of resources, or greater social isolation [9,48]. A greater rate of smoking has also been observed with other categories, especially for people who are separated [6,22]. For our sample, most of the non-married participants were divorced, and the smallest count was for separated people. These sample sizes might account for the differences in significance.

There are a number of caveats that should be mentioned. Foremost is that the final penalized regression model for our primary analysis had a weak overall effect. The set of variables in our final model did not discriminate very well among former and current smokers. Furthermore, some of the variables that were not selected for the final model (e.g., gender by divorced) had Z-values that might be considered non-trivial. Emphasis was on statistical significance to guard against over-inclusion in this large-sample and many-predictor analysis context. Some excluded variables might have practical impact yet were omitted because of redundancy with the retained variables. Because correlations among predictors are subject to sampling variability, it might be that the variables selected by our method could be different in a new sample. There is potentially a long list of variables outside our set that might account for more substantial differences in smoking behavior, and future research might include these. Our variable choice was limited by the constraints of the NLST, which was a study not primarily designed to address the research questions of our analysis. For example, indicators of socio-economic status other than education were not available.

Another caveat is that the NLST cohort is not necessarily representative of the US heavy smoking population. The sex ratio and pack years were consistent with the concurrent US Census Bureau findings, but the NLST cohort was younger, better educated and less likely to be currently smoking [58].

Finally, this study was cross-sectional and as such, does not provide information about within-person changes in smoking behavior over time. Our interpretations about changes in BMI and smoking behavior should be qualified, as it is unclear to what extent we can makes inference about temporal change based on between-person differences.

## 5. Conclusions

Our analysis indicates a fairly strong association of cg05575921 methylation and self-report smoking and provides specific information about the interaction of gender and body weight in predicting smoking behavior in an older cohort. Self-reported smoking appears to be reasonably accurate and the motivation to smoke might be similar across the lifespan. Perceptions about body weight might preclude the success of continued abstinence, especially for women.

## Figures and Tables

**Figure 1 genes-14-00025-f001:**
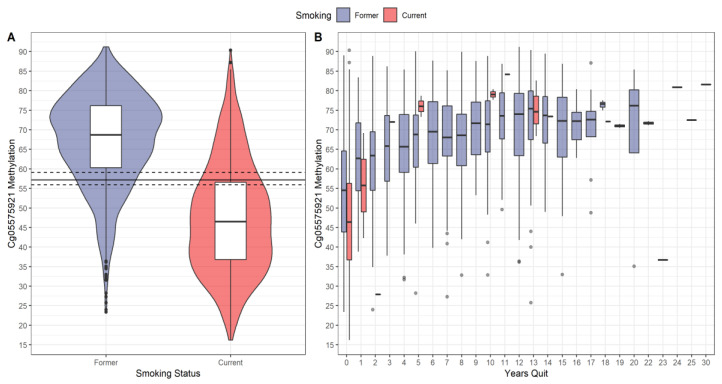
Distribution of cg05575921 methylation. (**A**) Boxplots wrapped by violin plots of cg05575921 methylation by self-reported smoking status. The solid horizonal line is the estimated optimal cut-point, and the dashed lines are the (asymmetric) 95% CI. (**B**) Boxplots of cg05575921 methylation as a function of smoking status and years quit (number of years since quitting cigarettes). A small number of self-reported current smokers had >0 years quit attributable to measurement error.

**Figure 2 genes-14-00025-f002:**
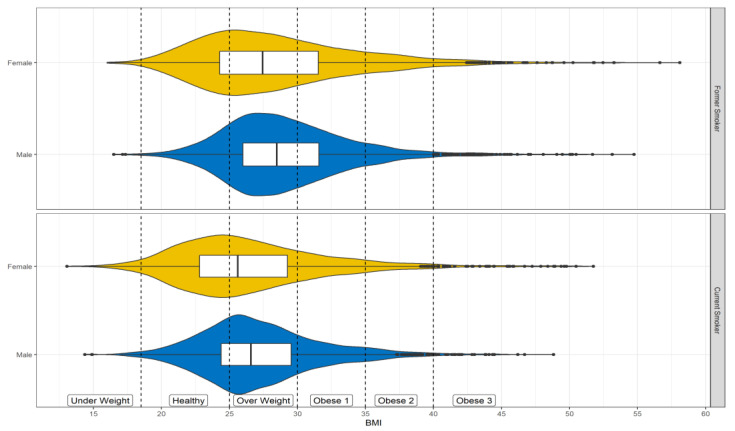
Distribution of body mass index (BMI) by gender and smoking status. Dashed lines and labels are the CDC classifications.

**Figure 3 genes-14-00025-f003:**
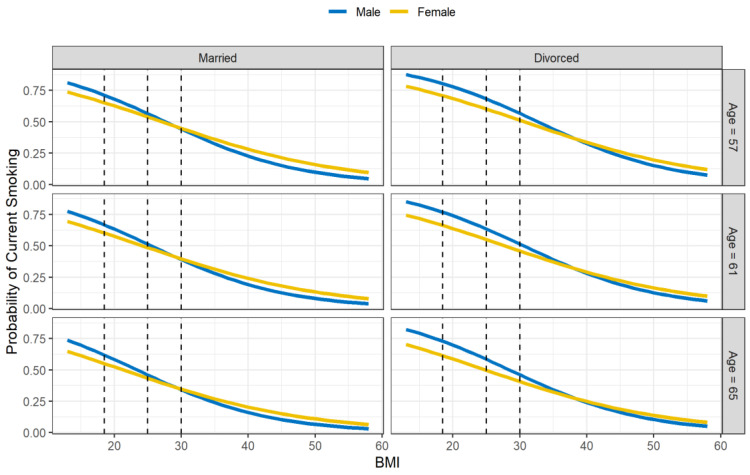
Probability of current smoking by BMI, gender, marital status (columns) and age (rows). Age values are the quartiles of the sample. Curves are predicted values based on the penalized logistic regression. Vertical dashed lines are the CDC classifications of underweight, healthy, overweight, and obese (left to right).

**Table 1 genes-14-00025-t001:** Descriptive statistics of key variables for three groupings of the overall cohort.

	With Methylation			Without Methylation			Combined		
Variable	*N*	Former Smoker, *N* = 1583 ^1^	Current Smoker, *N* = 1491 ^1^	*N*	Former Smoker, *N* = 3543 ^1^	Current Smoker, *N* = 3467 ^1^	*N*	Former Smoker, *N* = 5126 ^1^	Current Smoker, *N* = 4958 ^1^
Age	3074	61.96 (5.03)	61.18 (4.80)	7010	62.45 (5.22)	61.19 (5.03)	10,084	62.30 (5.17)	61.18 (4.96)
Female	3074	605 (38%)	651 (44%)	7010	1554 (44%)	1556 (45%)	10,084	2159 (42%)	2207 (45%)
White	2993	1501 (97%)	1373 (95%)	6851	3370 (97%)	3169 (94%)	9844	4871 (97%)	4542 (94%)
Educ > HS	3007	1135 (73%)	997 (68%)	6819	2516 (73%)	2242 (66%)	9826	3651 (73%)	3239 (67%)
Marital Status	3074			7010			10,084		
Never		84 (5.3%)	80 (5.4%)		187 (5.3%)	223 (6.4%)		271 (5.3%)	303 (6.1%)
Married		1104 (70%)	907 (61%)		2388 (67%)	2045 (59%)		3492 (68%)	2952 (60%)
Widowed		107 (6.8%)	121 (8.1%)		266 (7.5%)	301 (8.7%)		373 (7.3%)	422 (8.5%)
Separated		17 (1.1%)	21 (1.4%)		46 (1.3%)	62 (1.8%)		63 (1.2%)	83 (1.7%)
Divorced		271 (17%)	362 (24%)		656 (19%)	836 (24%)		927 (18%)	1198 (24%)
BMI	3065	28.80 (5.15)	26.98 (4.84)	6996	28.83 (5.15)	26.88 (4.93)	10,061	28.82 (5.15)	26.91 (4.90)
Pack Years	3074	56.49 (23.99)	54.80 (21.80)	7010	56.31 (24.48)	55.28 (21.96)	10,084	56.37 (24.33)	55.14 (21.91)
Cg05575921	3074	66.95 (12.14)	47.47 (14.03)	0	NA (NA)	NA (NA)	3074	66.95 (12.14)	47.47 (14.03)
Years Quit	3074	7.03 (5.06)	0.08 (0.97)	7010	7.00 (5.03)	0.04 (0.66)	10,084	7.01 (5.04)	0.05 (0.77)

^1^ Mean (SD) reported for quantitative variables; *N* (%) for categorical variables.

**Table 2 genes-14-00025-t002:** Results of the penalized logistic regression. Log odds ratios, z-values, and marginal false discovery rate (mFDR). The predictors selected for the final model are indicated by asterisks in the last column.

				Variable Selection	
Predictor	Estimate	Z	mFDR	Initial	Final
BMI	−0.099	−22.646	<0.001	*	*
Gender ^a^	−0.770	−18.005	<0.001	*	*
Gender by BMI	0.026	17.512	<0.001	*	*
Age	−0.053	−12.568	<0.001	*	*
Divorced ^b^	0.501	9.574	<0.001	*	*
Education ^c^	−0.345	−7.430	<0.001	*	*
Race ^d^	−0.710	−6.361	<0.001	*	*
Gender by Divorced	−0.244	−3.744	0.014	*	
Never Married	0.308	3.298	0.063	*	
Gender by Widowed	0.225	2.466	0.418	*	
Widowed	0.178	2.254	0.541	*	
Gender by Never	−0.213	−1.538	0.820	*	
Gender by Separated	0.439	1.361	0.855	*	
Separated	0.252	1.336	0.859	*	
Race by Education	0.000	0.047	0.937		

^a^ 0 = Male, 1 = Female; ^b^ 0 = Married, 1 = Divorced; ^c^ 0 = low, 1 = high; ^d^ 0 = African American, 1 = White.

## Data Availability

The clinical data can be accessed through the National Cancer Institute’s Cancer Data Access System (CDAS): https://cdas.cancer.gov/ (accessed on 9 May 2022). The cg05575921 methylation data included in this manuscript were prepared with funding from the United States National Institutes of Health (NIH). They are not yet publicly available.

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
