# Peer review of "Predictors of Smoking in Older Adults and an Epigenetic Validation of Self-Report"

_genes, 2022, doi:10.3390/genes14010025_

Round 1

Reviewer 1 Report

Dear Authors: I am glad to have the opportunity to review this work. I believe that the study contributes to the understanding of smoking behavior. Smoking cessation is challenging, due to the high relapse rates that exist in people who attempt it. Obtaining predictors from this large sample may help professionals engaged in smoking cessation treatment to decide where to focus interventions.

I have the following recommendations to improve the paper:

On page 3, line you should change the word "morality" to "mortality".

The link where you provide additional details of the National Lung Screening Trial (NLST), is not active.

On page 3, line 117, you mention that Cg05575921 methylation was determined using the Smoke Signature assay from Behavioural Diagnostics using methods previously described. However, I recommend making a brief description in the Methods section.

On page 5, lines 201 and 202, review effect values with log odds in current smokers for age; the values do not match those of table 2.

Although the correlation between cg05575921 methylation with cotinine levels would strengthen the study, the association with self-reporting is also valuable. You state that the final penalized regression model for our primary analysis had a weak overall effect. I consider it's important to state this because while some of the findings on methylation as an indicator of withdrawal have been encouraging, here it's clear that they should be taken with caution.

The conclusions point to one of the most important findings of the study, which is the interaction of BMI and gender as a predictor of smoking behavior. However, I suggest that in the conclusions you also state your position regarding cg05575921 methylation as a predictor of smoking behavior.

Author Response

Review 1

  1. On page 3, line x you should change the word "morality" to "mortality".

Typo is fixed on p. 3.

  1. The link where you provide additional details of the National Lung Screening Trial (NLST), is not active.

Apologies – the link now reflects the active NLST page on p. 3.

  1. On page 3, line 117, you mention that Cg05575921 methylation was determined using the Smoke Signature assay from Behavioural Diagnostics using methods previously described. However, I recommend making a brief description in the Methods section.

We agree with the suggestion. A brief description of the methylation method now appears on p. 3.

  1. On page 5, lines 201 and 202, review effect values with log odds in current smokers for age; the values do not match those of table 2.

Thanks for pointing this out. The numbers in the text now agree with those in Table 2 (see p.5).

  1. Although the correlation between cg05575921 methylation with cotinine levels would strengthen the study, the association with self-reporting is also valuable. You state that the final penalized regression model for our primary analysis had a weak overall effect. I consider it's important to state this because while some of the findings on methylation as an indicator of withdrawal have been encouraging, here it's clear that they should be taken with caution.

We agree with the reviewer. (We do not think this comment requires any changes to the manuscript.)

  1. The conclusions point to one of the most important findings of the study, which is the interaction of BMI and gender as a predictor of smoking behavior. However, I suggest that in the conclusions you also state your position regarding cg05575921 methylation as a predictor of smoking behavior.

We appreciate the reviewer’s point, and we now state our position regarding cg05575921 methylation in the conclusion on p. 9.

Reviewer 2 Report

The research aims to characterise successful smoking cessation among older, long-term smokers in terms of their characteristics, e.g. age, BMI, marital status, race, etc. This is achieved by applying penalised regression to clinical data collected from the National Lung Screening Trial to derive a linear of characteristics that best predicts self-report smoking status. Consistent with previous literature, the resulting model implies that smoking cessation is higher in individuals who are married, white, higher educated, have higher BMI, and are older. Self report smoking status is compared to DNA methylation of the cg05575921 CpG site which is well-known to be less methylated in current smokers but slowly reverts to non-smoking levels in a few years following smoking cessation.

Major comments

As NLST recruited individuals of a specific profile, there should be some discussion of the potential influence of this selection bias on findings and why there was no effort to to mitigate potential effects.

The implications of the model reported in Table 2 are not adequately discussed. First, discussions rely almost entirely on statistical significance. However, it is possible for an important predictor to lack statistical significance for at least two reasons: (1) low power and (2) high correlation with other predictors. Neither of these possibilities are mentioned or investigated.  Second, even statistically significant predictors can be irrelevant for clinical practice or improving public health if the effect is small.  Although effect estimates are reported, their practical impact is not described or discussed.

The utility of DNA methylation in the study is minor. It is claimed to validate smoking status self report. If it is a proxy for smoking status, why is it not also used to validate associations with smoking status?

Minor comments

Reference to "smoking behavior" in the title is too general. The study is specifically and only investigating smoking cessation.

Line 98 'morality' -> 'mortality'

Section 3.1 is describes self-report smoking validation poorly. First, it says that "optimal cut-point analysis" produced an AUC. AUC values are produced using AUROC analysis. Cut-point analysis is applied to ROC curves to identify a threshold that optimizes some threshold. Second, it doesn't say what metric is being optimized by the cut-point analysis. Third, it should give the units of the cut-point, i.e. that it is 57.19% DNA methylation of CpG site cg05575921.

Caption for Figure 1A: "solid vertical line" -> "solid horizontal line"

Figure 1B is confusing. In particular, how is it possible for "current smokers" to have a value for "years quit"? If they are current smokers, then they haven't quit.

Line 206 should make it clear that the performances denoted by AUC and R^2 were obtained in the same data in which the model was derived.  As a result, these metrics are almost certainly inflated.

The discussion refers to Figure 1 as evidence that there is some misreporting in self report smoking (see "Despite some probable misreporting (see Figure 1) ..."). It's unclear why this might be the case.

The 'timing of the transition to the greater rate of female smoking' assumes that male smoking is constant across BMI. In fact, male smoking decreases with BMI, consequently it is unclear what meaning can be assigned to this 'timing'.

The discussion contains the statement: "The prospect that women tend to continue smoking longer than men is a public health concern." What is the evidence for this? 

Author Response

Reviewer 2

  1. (Major comment). As NLST recruited individuals of a specific profile, there should be some discussion of the potential influence of this selection bias on findings and why there was no effort to mitigate potential effects.

The reviewer raises an important point. On p. 9 we now mention similarities and differences of the NLST cohort with the concurrent US Census Bureau findings. We do agree with the reviewer that the NLST is a biased sample. Our intention was not to remedy the bias, but to provide findings for this large cohort and compare our results with other cohorts.  

  1. (Major comment). The implications of the model reported in Table 2 are not adequately discussed. First, discussions rely almost entirely on statistical significance. However, it is possible for an important predictor to lack statistical significance for at least two reasons: (1) low power and (2) high correlation with other predictors. Neither of these possibilities are mentioned or investigated. Second, even statistically significant predictors can be irrelevant for clinical practice or improving public health if the effect is small.  Although effect estimates are reported, their practical impact is not described or discussed.

We agree with several of the reviewer’s points and now address them on p. 9. Regarding statistical significance, we note on p. 4 that this is a large-N and moderate-P (predictors) analysis. The intent was to guard against over-inclusion of variables, which was accomplished using the penalized regression in concert with the marginal false discovery rate. The reviewer is correct that some of the excluded effects might have practical importance, which we now acknowledge in the added material on p. 9.   

  1. (Major comment). The utility of DNA methylation in the study is minor. It is claimed to validate smoking status self-report. If it is a proxy for smoking status, why is it not also used to validate associations with smoking status?

An aspect of the reviewer’s comment that we agree with is that the methylation was secondary to the primary analysis of predicting smoking behavior. Nevertheless, as mentioned on p. 2, it is uncommon to validate the self-report that is usually the prime object of analysis. Because there are some studies that show self-report can be inaccurate, we took the opportunity to provide validity evidence for the self-report. Having shown evidence that the self-report was fairly accurate, we wanted to use the full sample available to us for the predictor analysis.

  1. (Minor comment). Reference to "smoking behavior" in the title is too general. The study is specifically and only investigating smoking cessation.

There is agreement that “smoking behavior” is too general. But we think this is due to using the word “behavior”. For this reason, we now omit “behavior” from the title.

  1. (Minor comment). Line 98 'morality' -> 'mortality'

This is now corrected (see p. 3).

  1. (Minor comment). Section 3.1 describes self-report smoking validation poorly. First, it says that "optimal cut-point analysis" produced an AUC. AUC values are produced using AUROC analysis. Cut-point analysis is applied to ROC curves to identify a threshold that optimizes some threshold. Second, it doesn't say what metric is being optimized by the cut-point analysis. Third, it should give the units of the cut-point, i.e. that it is 57.19% DNA methylation of CpG site cg05575921.

It is agreed that the description of the cut-point analysis can be improved. We have revised the discussion on p. 5 along the lines suggested by the reviewer. The details of AUC and AUROC are discussed on p. 3, consistent with the reviewer’s comments. We chose to omit “%” for reported methylation values for simplicity’s sake. The new material on p. 3 does clearly indicate that methylation is a percentage.

  1. (Minor comment). Caption for Figure 1A: "solid vertical line" -> "solid horizontal line"

This is now corrected in the Figure 1A caption.

  1. (Minor comment). Figure 1B is confusing. In particular, how is it possible for "current smokers" to have a value for "years quit"? If they are current smokers, then they haven't quit.

The phenomenon that the reviewer mentions is noted on p. 4 and shown in Table 1. The most likely explanation of current smokers reporting > 0 years quit is measurement error. That is, either the status (former/current) is misreported/mis-recorded, or the years quit is misreported/mis-recorded. We now add clarification on p. 5 and the Figure 1B caption.  

  1. (Minor comment). Line 206 should make it clear that the performances denoted by AUC and R^2 were obtained in the same data in which the model was derived. As a result, these metrics are almost certainly inflated.

We used bootstrap cross-validation for the performance indices. This is explained on p. 4, but now reiterated on p. 6. The method is better than computing the statistics on the entire sample, but we agree with the reviewer that external validation in a new sample is ideal.

  1. (Minor comment). The discussion refers to Figure 1 as evidence that there is some misreporting in self report smoking (see "Despite some probable misreporting (see Figure 1) ..."). It's unclear why this might be the case.

Please see our response to comment #8.

  1. (Minor comment). The 'timing of the transition to the greater rate of female smoking' assumes that male smoking is constant across BMI. In fact, male smoking decreases with BMI, consequently it is unclear what meaning can be assigned to this 'timing'.

As the reviewer points out, the term “timing” is potentially misleading and is now omitted (p. 9).

  1. (Minor comment). The discussion contains the statement: "The prospect that women tend to continue smoking longer than men is a public health concern." What is the evidence for this?

The reviewer refers to a sentence on p. 9 that sets up the evidential statements that follow. The opening sentence is not supported by a citation, whereas the evidential statements are. For this reason, we now omit the opening sentence and let the following statements stand on their own (with citations).  

Round 2

Reviewer 2 Report

Thank you for responses. However, a couple of issues remain:

1. The statement on page 5 is still incorrect:

"The cut-point analysis for finding the optimal cg05575921 methylation value resulted 189 in AUC = 0.85 with 95% CI = [0.83, 0.86]."

Cut-point analysis does not result in an AUC value, it calculates an optimal cut-point. In fact, the statement on page contradicts the methods related to AUC and cut-point given on page 3.

2. The statement added to the discussion in response to my review is incorrect and inadequate ("Furthermore, some of the effects ..."):

(a) variables are included in a model, not 'effects'

(b)  the response perpetuates a critical misunderstanding about penalised regression. It assumes that the model generated includes the variables individually most robustly associated with the outcome (in this case, smoking).  This is not the case.  These would be identified by running association tests each focused on the relationship between an individual variable and the outcome.  Penalised regression aims to construct a model from multiple variables that best explains variance in the outcome. As a result, some variables with robust associations may be excluded because they do not contribute to this explanation beyond a selected model.  Similarly, some variables with very weak associations with the outcome may be included in the model because it explains some variance missed by the other variables that were included. It is possible that if your dataset were slightly different, e.g. a few individuals removed or additional individuals included, that your final model would not include exactly the same variables. 

Author Response

Our responses are in italic font.

Reviewer

  1. The statement on page 5 is still incorrect:

"The cut-point analysis for finding the optimal cg05575921 methylation value resulted 189 in AUC = 0.85 with 95% CI = [0.83, 0.86]."

Cut-point analysis does not result in an AUC value, it calculates an optimal cut-point. In fact, the statement on page contradicts the methods related to AUC and cut-point given on page 3.

We agree that the language needs to be clarified. We now identify the ROC analysis on p. 3 and state on p. 5 that “The ROC analysis resulted in the AUC = .85…”

  1. The statement added to the discussion in response to my review is incorrect and inadequate ("Furthermore, some of the effects ..."):

(a) variables are included in a model, not 'effects'

On p. 9 we now use the term “variables” rather than “effects”.

(b)  the response perpetuates a critical misunderstanding about penalised regression. It assumes that the model generated includes the variables individually most robustly associated with the outcome (in this case, smoking).  This is not the case.  These would be identified by running association tests each focused on the relationship between an individual variable and the outcome.  Penalised regression aims to construct a model from multiple variables that best explains variance in the outcome. As a result, some variables with robust associations may be excluded because they do not contribute to this explanation beyond a selected model.  Similarly, some variables with very weak associations with the outcome may be included in the model because it explains some variance missed by the other variables that were included. It is possible that if your dataset were slightly different, e.g. a few individuals removed or additional individuals included, that your final model would not include exactly the same variables.

The goal of MCP regression is to eliminate the unimportant predictors from the model while leaving the important predictors unpenalized. This is equivalent to fitting an unpenalized model in which the truly nonzero predictors are known beforehand. Thus, the non-zero regression coefficients in the penalized model have similar interpretations as in traditional regression, meaning they adjust for the correlations among the variables. We agree with the reviewer that the Z-value does not represent an unadjusted or univariate effect. But with respect, we do not see where our comments in any way suggest that the penalized regression coefficients should be interpreted in an unadjusted or univariate fashion. Furthermore, we fail to see how our comments on p. 9 are perpetuating misunderstanding about penalized regression; our comment regarding interpretation of the size of a Z-value is simply read off Table 2. We agree with the reviewer that sampling variability will affect variable selection. On p. 9 we now add the following comments to cover both issues:

“Some excluded variables might have practical impact yet were omitted because of redundancy with the retained variables. Because correlations among predictors are subject to sampling variability, it might be that the variables selected by our method could be different in a new sample.”